# Palm Kernel Meal Protein Hydrolysates Enhance Post-Thawed Boar Sperm Quality

**DOI:** 10.3390/ani13193040

**Published:** 2023-09-27

**Authors:** Vassakorn Khophloiklang, Panida Chanapiwat, Ratchaneewan Aunpad, Kampon Kaeoket

**Affiliations:** 1Semen Laboratory, Department of Clinical Sciences and Public Health, Faculty of Veterinary Science, Mahidol University, Nakhon Pathom 73170, Thailand; vassakorn.khp@student.mahidol.edu (V.K.); panida.chn@mahidol.edu (P.C.); 2Faculty of Veterinary Science, Rajamangala University of Technology Srivijaya, Nakhon Si Thammarat 80240, Thailand; 3Graduate Program in Biomedical Sciences, Faculty of Allied Health Sciences, Thammasat University, Pathum Thani 12120, Thailand; aunpad@gmail.com

**Keywords:** antioxidant, boar sperm, freezing, palm kernel meal protein hydrolysates, peptides

## Abstract

**Simple Summary:**

Semen cryopreservation is associated with the production of reactive oxygen species, which leads to lipid peroxidation of sperm membranes, resulting in a reduction in sperm motility and decreased fertilizing ability. We investigated the effects of bioactive peptides from palm kernel meal protein hydrolysates (PKMPHs) on sperm viability, motility, acrosome and mitochondrial function in frozen–thawed boar semen. PKMPH improved the total motility, progressive motility, viability, acrosome integrity, and mitochondrial function of sperm and diminished lipid peroxidation. Therefore, we suggest that PKMPH plays a crucial role as an antioxidant during the freezing of boar sperm.

**Abstract:**

Boar sperm is sensitive to particular conditions during cryopreservation, resulting in an extreme reduction in fertilizing ability due to damage to the sperm membranes. PKMPH contains bioactive peptides that have antioxidant and antimicrobial activities. There is no information on the use of palm-kernel-meal-derived bioactive peptides for boar semen cryopreservation. This study aimed to examine the effects of bioactive peptides from PKMPH on post-thawed boar sperm quality. Boar semen ejaculates (*n* = 17) were collected and divided into six equal aliquots based on PKMPH concentrations (0, 1.25, 2.5, 5, 10, and 15 µg/mL) in a freezing extender. Semen samples were processed and cryopreserved using the liquid nitrogen vapor method. Thereafter, the frozen semen samples were thawed at 50 °C for 12 s and evaluated for sperm motility using a computer-assisted sperm analyzer and for sperm viability, acrosome integrity, mitochondrial function, and lipid peroxidation by measuring the level of malondialdehyde (MDA). The results demonstrate that the supplementation of PKMPH with 2.5 µg/mL afforded superior post-thawed sperm qualities, such as increased total motility, viability, acrosome integrity, and mitochondrial function by 10.7%, 12.3%, 18.3%, and 12.7%, respectively, when compared to the control group. PKMPH at a concentration of 2.5 µg/mL showed the lowest level of MDA (40.6 ± 2.0 µMol/L) compared to the other groups. In conclusion, adding PKMPH peptides at 2.5 µg/mL to the freezing extender reduced the oxidative damage associated with cryopreservation and resulted in higher post-thawed sperm quality.

## 1. Introduction

Boar sperm is sensitive to particular conditions during cryopreservation, including oxidative damage, intracellular and extracellular ice formation during the semen cryopreservation and thawing processes, excessive formation of reactive oxygen species (ROS), and osmotic stress [1]. It is worth noting that cryopreserving boar semen results in an extreme reduction in fertilizing ability due to damage to the sperm membranes. Since mammalian sperm contains a high proportion of polyunsaturated fatty acids (PUFAs), the membrane lipid composition of spermatozoa is correlated to specific functions with different fluidity and permeability characteristics necessary for reaching and fusing with the oocyte [2]. Additionally, the lipid composition of the sperm plasma membrane is a major determinant of its mobility characteristics, cold sensitivity, overall viability, and membrane integrity. Most boar sperm contains a higher proportion of PUFAs in the plasma membrane than other species [3]. It has been documented that PUFAs play a key role in sperm membranes’ fluidity and susceptibility to lipid peroxidation. Oxidative stress and lipid peroxidation during cryopreservation cause sperm dysfunction associated with decreased membrane fluidity following the loss of membrane integrity, which is essential for sperm function [4]. To protect sperm cells from cryopreservation damage, supplementation with a robust defense of antioxidants could protect against the oxidative stress induced by free radicals. In the past ten years, significant emphasis has been placed on investigating the beneficial impact of antioxidants on both the quality of cryopreserved sperm and the fertilizing capacity of boar sperm and other species, such as vitamin E [5] and glutathione [6]. Amino acids such as L-cysteine [7], N-acetyl L-cysteine [8], methionine [9], and arginine [10] have been examined, along with plant-based antioxidants such as gamma-oryzanol [11], quercetin [12], curcumin [13], and other polyphenols from plant extracts, which showed protection effects for sperm cryopreservation from boars and others [14].

Until now, palm kernel meal protein hydrolysates (PKMPH) have gained much attention not only because this digested protein has antimicrobial properties but also because of its antioxidant properties [15,16]. PKMPH, with its high protein content and antimicrobial peptides, can be used directly as feed supplementation for nursery pigs in order to promote their growth rate and health [17]. It has been shown that protein hydrolysate consists of a complex mixture of peptides and amino acids, which have important biological activities such as antioxidant, antihypertensive, antimicrobial, and anticoagulant activities [18,19]. In particular, amino acids with high antioxidant capacity mostly have (nucleophilic) sulfur-containing side chains, such as cysteine and methionine, or easily transferable hydrogen atoms [20]. Many in vitro studies reported that the antioxidant activity of protein hydrolysate depends on its peptides and amino acid compositions [18,20,21,22,23]. However, there have been no studies reported on the antioxidant effects of PKMPH during the cryopreservation of boar semen. Therefore, this study aims to elucidate the cryoprotection effect of PKMPH by evaluating the qualities of boar semen post thaw. 

## 2. Materials and Methods

### 2.1. Animal

Semen ejaculates (*n* = 17) were collected from a commercial pig farm from 10 boars, including the Landrace, Large White, and Duroc breeds (their ages ranged from 1.5 to 3 years old). They were kept in individual pens in an evaporative cooling system. Fresh, clean water was provided ad libitum with automated watering; the feeding level was adjusted to meet the semen production requirements of 3 kg/day. Boars were used for routine semen collection for artificial insemination.

### 2.2. Chemicals and Extenders

To prepare the PKMPH stock solution for the experiment, PKMPH powder (produced from palm kernel meal; patent registered number 2203001859) was diluted in distilled water at 5 mg/mL. In this study, there were three extenders used for boar semen cryopreservation, as follows: Extender I was a commercial semen extender for the Beltsville Thawing Solution (BTS, Minitube, Tiefenbach, Germany). Extender II was composed of 20% egg yolk and 11% lactose solution supplemented with different concentrations of PKMPH (0, 1.25, 2.5, 5, 10, and 15 µg/mL) in distilled water. Each aliquot was diluted with or without (control) PKMPH substance with an equal extender volume. Extender III was composed of 89.5% extender II, with 9% (*v*/*v*) glycerol and 1.5% (*v*/*v*) Equex-STM^®^ (Nova Chemical Sales Inc., Scituate, MA, USA).

### 2.3. Semen Collection and Preparation

Boar semen samples were collected using the gloved-hand technique. The semen was filtered through gauze, and only sperm-rich samples were collected and evaluated with parameters including semen volume, pH, sperm motility, concentration, sperm viability, and morphology. The semen samples with a motility of >70% and that were 80% morphologically normal were used for cryopreservation [24]. 

### 2.4. Semen-Freezing Process

All samples of sperm were frozen using the traditional nitrogen technique. In brief, the semen was diluted (1:1 *v*/*v*) with the BTS extender shortly after collection. The diluted semen was transferred to 50 mL centrifuge tubes and equilibrated at 15 °C for 120 min before being centrifuged at 15 °C at 800× *g* for 10 min (LMC-4200R, Biosan, Riga, Latvia). After centrifugation, the supernatant was discarded, and the sperm pellet was re-suspended (approximately 1–2:1) in Extender II at a concentration of 1.5 × 10^9^ spermatozoa/mL [11]. At this step, the sperm sample was divided into six groups based on the PKMPH concentration (0, 1.25, 2.5, 5, 10, and 15 µg/mL in distilled water). All of the sperm samples were chilled to 5 °C for 90 min. Each sample group was mixed with Extender III to a concentration of 1.0 × 10^9^ sperm/mL. The processed semen was loaded into 0.5 mL straws (IMV Technologies, L’Aigle, Basse-Normandie, France). The straws were placed in contact with nitrogen vapor (4 cm above the level of the liquid nitrogen) for 20 min (−20 °C/min) in a polystyrene box and plunged into the liquid nitrogen tank (−196 °C) for storage prior to analysis [11]. The frozen semen was kept for 12 h before evaluation.

### 2.5. Thawing Process

Before sperm evaluation, the frozen semen samples were thawed at 50 °C for 12 s and extended (1:6) with a pre-warmed BTS extender at 37 °C for 15 min [11].

### 2.6. Assessment of Sperm Motility

The sperm motility was conducted using computer-assisted sperm motility analysis (CASA) (AndroVision^®^, Minitube, Tiefenbach, Germany). Briefly, 3 µL of semen sample was carefully transferred into a disposable counting chamber (Leja^®^ 20 µM, IMV Technologies, L’Aigle, Basse-Normandie, France) and maintained at 37 °C throughout the analysis. A minimum of 600 sperm cells were counted in each analysis by examining five fields from each sample. The results were represented as the percentage of total sperm motility, progressive motility, and motility patterns, including curvilinear velocity (VCL, µm/s), average pathway velocity (VAP, mm/s), straight-line velocity (VSL, mm/s), amplitude of lateral head displacement (ALH, mm), straightness (STR, %), and linearity (LIN, %). Motile spermatozoa were defined with VCL ≥ 24 µm/s and ALH > 1 µm. Progressive motility (PMOT) was interpreted as the presence of a VCL ≥ 48 µm/s and a VSL < 10 µm/s. The total motility (MOT) is the summation of sperm motility subpopulations that were determined by VCL thresholds, including local motility (VCL ≥ 24 and < 48 µm/s), slow motility (VCL ≥ 48 and < 80 µm/s), and fast motility (VCL ≥ 80 µm/s) [25].

### 2.7. Assessment of Sperm Morphology

Sperm morphology was evaluated using William’s staining method (carbolfuchsin-eosin) under a light microscope at 400× magnification, and the results were expressed as the percentage of normal spermatozoa [13].

### 2.8. Assessment of Sperm Viability

The sperm viability was assessed using the staining techniques of SYBR-14 (Sperm viability kit, Molecular Probes, L7011) and Ethidiumhomodimer-1 (EthD-1). Briefly, 10 µL of semen was combined with 2.7 µL of SYBR-14 (0.54 µM in DMSO) and 10 µL of 1.17 µM EthD-1. The mixture was incubated at 37 °C for 15 min. Under a 400× magnification fluorescence microscope, 200 sperm were assessed. The SYBR-14/EhtD-1-stained sperm were classified into viable and non-viable sperm. The nuclei of live sperm with an intact plasma membrane fluoresced green, whereas the nuclei of dead sperm or damaged plasma membranes fluoresced red (Figure 1). The percentage of viable and non-viable sperm was calculated [11,13].

### 2.9. Assessment of Acrosome Integrity

The acrosome integrity was assessed with fluorescein isothiocyanate-labeled peanut (*Arachis hypogaea*) agglutinin (FITC-PNA) staining. A 10 µL specimen of diluted semen was mixed with 10 µL of 1.17 µM EthD-1 and incubated at 37 °C for 15 min. Then, 5 µL of the sample was smeared onto a glass slide and allowed to air dry. The sample was fixed with 95% ethanol for 30 s and air-dried. Then, 40 µL of FITC-PNA (100 µg/mL in PBS) was spread over the slides and incubated in a moist chamber at 4 °C for 30 min. The slides were rinsed with cold PBS and air-dried. Under a 1000× magnification fluorescence microscope, 200 sperm were evaluated, and the sperm with intact acrosomes were shown in green color with a smooth contour in the acrosomal region, while the damaged acrosomes were shown in green with a rough contour (Figure 1). The results were expressed as a percentage of intact sperm acrosomes [11,13].

### 2.10. Assessment of Mitochondrial Membrane Potential

The mitochondrial membrane potential of the sperm was evaluated by fluorescent multiple staining using propidium iodide (PI; 0.5 mg/mL), FITC-PNA (1 mg/mL in PBS), and 5,5’,6,6’-tetrachloro-1,1′,3,3′-tetraethylbenzimidazolylcarbocyanine iodide (JC-1 (1.53 mM) diluted with DMSO (ratio 1:10) (T3168, Invitrogen, Waltham, MA, USA) staining. Then, 150 μL of frozen–thawed semen was mixed with 2.5 μL of PI and incubated at 37 °C for 5 min. Then, the mitochondria were labeled using 2 μL of JC-1 and incubated at 37 °C for 10 min. Finally, the acrosomes were labeled using 2 μL of FITC-PNA and incubated at 37 °C for 15 min in the dark [26]. Under a 400× magnification fluorescence microscope, 200 sperm were assessed. Midpiece staining showed that sperm with a high mitochondrial membrane potential fluoresced yellow-orange, whereas sperm with a low membrane potential fluoresced green (Figure 1). The percentage of sperm with high mitochondrial membrane potential was calculated [27].

### 2.11. Assessment of Lipid Peroxidation

Lipid peroxidation was evaluated by measuring the concentration of malondialdehyde (MDA) using a colorimetric lipid peroxidation assay kit (ab118970, Abcam^®^, Cambridge, UK) following the manufacturer’s instructions. Briefly, 250 µL of post-thawed sperm in each treatment was diluted with BTS (1:4), centrifuged at 1300× *g* for 5 min, the supernatant was removed, and the sperm pellet was resuspended with cold PBS (repeated twice). The sperm pellet was lysed by lysis buffer and homogenizer on ice and then centrifuged at 13,000× *g* for 10 min. The supernatant was collected and analyzed. The MDA in the supernatant sample (200 µL) reacted with 600 µL of thiobarbituric acid (TBA) solution to generate an MDA-TBA adduct that was quantified. The MDA-TBA product was measured immediately in a microplate reader (SPECTROstar Nano, BMG LABTECH, Ortenberg, Germany) at 532 nm. For the colorimetric assay, a 2 mM MDA standard was prepared and serially diluted for the standard curve. The levels of MDA were calculated from the MDA standard curve and expressed as µMol/L.

### 2.12. Statistical Analysis

Statistical analysis was performed using IBM SPSS Statistics for Windows, version 26.0 (SPSS Inc., Chicago, IL, USA). The normal distribution test of the data was examined using the Shapiro–Wilk test and the parameters included total motility, progressive motility, sperm motility patterns, sperm viability, acrosome integrity, mitochondrial membrane potential, and MDA, which were presented as mean ± SEM. The means were analyzed with a one-way ANOVA according to Randomized Complete Block Design (RCBD). Each treatment (different concentrations of PKMPH in a freezing extender) was the fixed factor, and the boars were the random block factor. The comparison of sperm parameters among treatment groups was performed by Duncan’s multiple-range test. A statistically significant difference was defined as *p* < 0.05.

## 3. Results

### 3.1. Effects of PKMPH on Sperm Motility

Table 1 displays the descriptive statistics concerning the quality of fresh boar semen. All measurements confirm that the quality of the fresh semen samples was acceptable. The effects of palm kernel meal protein hydrolysates on sperm motility are shown in Figure 2. The results demonstrated that the supplementation of PKMPH with 2.5 µg/mL showed superior post-thawed sperm quality compared to other concentrations. Semen samples supplemented with 2.5 µg/mL improved motility by 10.7% when compared with the control (38.3 ± 2.3% vs. 27.6 ± 1.8%). There was a significant difference in the treatment groups at 2.5 and 5 µg/mL when compared with the control group in terms of sperm motility characteristics (P-MOT, VCL, VSL, VAP, ALH, STR, and LIN), as evaluated using a CASA (Table 2).

### 3.2. Effects of PKMPH on Sperm Viability

The results of sperm viability in the treatment groups were higher than in the control group, as shown in Figure 3. PKMPH at a concentration of 2.5 µg/mL showed the highest number of viable sperm (40.5 ± 1.8%), which was 12.3% higher than the control group.

### 3.3. Effects of PKMPH on Sperm Acrosome Integrity

A variation in acrosome integrity was observed among the control and treatment groups (Figure 4). However, the highest percentage of acrosome integrity was found at a concentration of 2.5 µg/mL (51.4 *±* 2.2%), which was higher than the control group by 18.2%.

### 3.4. Effects of PKMPH on Mitochondrial Membrane Potential

The post-thawed sperm mitochondrial membrane potential is presented in Figure 5. The higher percentage of mitochondrial function in the group supplemented with 2.5 µg/mL (40.2 *±* 1.6%) was found when compared with the other groups.

### 3.5. Effects of PKMPH on Lipid Peroxidation

The effect of PKMPH on lipid peroxidation during cryopreservation is shown in Figure 6. There was a tendency for inferior MDA levels in the treatment groups than in the control group (*p* = 0.2). However, PKMPH at a concentration of 2.5 µg/mL showed the lowest level of MDA (40.6 ± 2.0 µMol/L) compared with the other groups.

## 4. Discussion

During the process of mammalian sperm cryopreservation, cryoinjuries (i.e., freezing and thawing processes) of sperm cause oxidation stress and the overproduction of ROS, which leads to an imbalance of the antioxidant system (i.e., antioxidant abilities versus excessive production of ROS), subsequently affecting semen qualities such as sperm motility, mitochondrial activity, membrane permeability, sperm functions, and survival rates [1,28]. 

To our knowledge, there is no information on using PKMPH, which contains bioactive peptides, for boar semen cryopreservation. This is the first study to observe the effect of PKMPH and its optimal concentration on frozen boar semen qualities. The present results clearly showed that adding PKMPH to the freezing extender significantly improved the post-thawed boar semen parameters, including total motility (10.7%), progressive motility (7.8%), sperm viability (12.3%), acrosome integrity (18.2%), mitochondrial membrane potential (12.7%), and kinetic motility patterns such as VCL, VSL, VAP, ALH, and LIN. Inferior MDA levels, indicating lower lipid peroxidation, were found in the treatment groups compared to the control, particularly when supplemented with PKMPH at a concentration of 2.5 µg/mL. This might be explained by the fact that spermatozoa absorbed and utilized bioactive peptides from the freezing extender to neutralize ROS, which, in turn, reduced lipid peroxidation in the plasma membrane and inner organelles [29,30]. A high variation of results in each parameter found in the present study may be explained by individual boar variation in terms of the good and poor freezability of each boar sperm, such as differences in sperm biochemical characteristics, seminal plasma composition and physiology, the polymorphism of the testis and epididymis, and PUFAs in the plasma membrane, which are linked to cryo-preservability, which was previously described by Yeste [28]. Taking all of these results together, the addition of 2.5 µg/mL of PKMPH is considered the optimal concentration for boar semen cryopreservation. This indicated that PKMPH-containing peptides with their antioxidant activity have protective effects on spermatozoa during freezing, thus reducing the degree of damage to their membrane, acrosome, and mitochondria. In agreement with our study, it has been reported that BAPT, bioactive peptides from the spleen of small-tailed Han sheep with short sequences of amino acids, improve total motility, live sperm, and intact acrosomes and protect ram sperm during cryopreservation [30]. In addition, feeding peptides from fermented soybean meals (FSBM) in pigs showed antioxidant activity by increasing serum antioxidant capacity [17,31], superoxide dismutase (SOD), and glutathione peroxidase (GSH-Px) activity [17], but decreased MDA concentrations [17,31]. Besides antioxidant activity, this FSBM also has beneficial effects on growth performance, immunity, digestibility, and intestinal morphology [17,31]. According to reports by Chanapiwat and Kaeoket [13] and Kaeoket et al. [11], plant-based antioxidants such as curcumin, gamma-oryzanol, and resveratrol improved post-thawed boar semen quality in terms of plasma membrane integrity and acrosome integrity. In this study, PKMPH, a plant-based peptide, also revealed antioxidant capacity by protecting sperm from the cryodamage associated with sperm freezing and thawing. However, too high a concentration of PKMPH showed a slightly cytotoxic effect on sperm qualities, which has also been reported for other antioxidants [32,33]. The reason might be that an excess of antioxidants may disturb the balance between free radicals and antioxidants in mammalian cells [34]. However, the underlying mechanism of PKMPH as an antioxidant for the cryoprotection of boar semen needs further study.

It has been documented that the antioxidant activity of protein hydrolysate depends on its peptide and amino acid compositions [20]. There are several pathways through which proteins or peptides could act as an antioxidative substance, such as the inhibition of singlet oxygen, causing lipid peroxidation and DNA damage through constitutive antioxidant defense systems such as free radical scavenging and the chelation of prooxidative transition metals, which have been reported to involve the -NH group of the imidazole ring and also inhibit free radical reactions through their proton-donating capability [20,21]. The PKMPH used in this study, consisting of nine peptides, contains 9–11 amino acids as follows: VDEVLNAPREE (P1), FFDEESFLH (P2), AGITDYFDED (P3), EADRTDYPE (P4), ISDETIDAIH (P5), LRPPSEEEE (P6), VLRPPSEEEE (P7), READSDDYPE (P8), and SFDQPAREVDE (P9) [21]. Most antioxidant peptides from PKMPH comprise basic amino acids such as Arg (R) and His (H), which are positively charged amino acid groups. These amino acid groups can donate protons or electrons to free radical molecules, and all peptide sequences contain hydrophobic amino acids, including Ala (A), Val (V), Ile (I), and Leu (L) [20]. Increasing hydrophobicity may result in increased lipid solubility and antioxidative activity [20,21]. In an in vitro study, Surangkulwattana et al. [21] demonstrated that PKMPH has high antioxidant activity, i.e., showing DPPH radical scavenging with IC_50_ values of 5.73 ± 0.23 μg/mL (PKMPH) versus 28.25 ± 0.97 μg/mL (L-glutathione), and ABTS radical scavenging activity of 7.84 ± 0.89 μg/mL (PKMPH) versus 14.03 ± 0.58 μg/mL (L-glutathione). In this study, we utilized PKMPH, whose function as an antioxidant was previously elucidated by Surangkulwattana et al. [21]. Besides the above function, the mechanism and pathway of bioactive peptides from PKMPH could be that they can stimulate antioxidant enzymes including SOD, GSH-Px, and catalase [22]. However, it is worth noting that an excess of antioxidants can interfere with redox signaling pathways, potentially disrupting normal cell function, as the suppression of antioxidant enzyme genes or high antioxidant levels can disrupt cellular homeostasis balance, leading to potential negative consequences for cellular health [35,36]. This might be explained by the present results in that adding the optimal PKMPH at 2.5 µg/mL in the freezing extender showed a superior post-thawed quality; in other words, upon adding a concentration of more than 5 g/mL, an inferior post-thawed semen quality can be seen. However, further studies on the fertility test on pig farms using PKMPH supplemented with frozen–thawed sperm for artificial insemination would add interest and significant credibility to the swine industry to maximize fertility rates. 

## 5. Conclusions

From these results, it can be concluded that PKMPH, with its bioactive peptide, is capable of reducing ROS generation during cryopreservation, inhibiting lipid peroxidation, and improving sperm motility, viability, intact acrosomes, and the mitochondrial membrane potential of frozen–thawed boar sperm. The PKMPH supplementation at an optimal dose of 2.5 µg/mL in a freezing extender improves the post-thawed boar semen quality. However, the supplementation of more than 5 µg/mL could lower the post-thawed semen quality.

## Figures and Tables

**Figure 1 animals-13-03040-f001:**
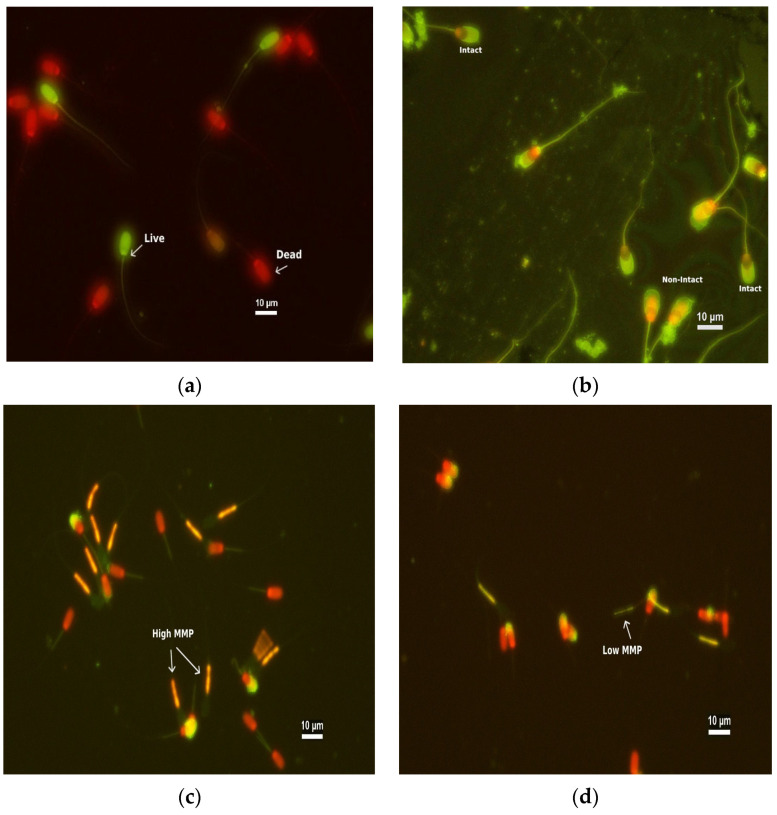
Assessment of frozen–thawed boar spermatozoa using specific fluorescent dyes (400× magnification): (**a**) Viable (live) and non-viable (dead) sperm stained with SYBR-14 (green) and EthD-1 (red); (**b**) Intact and non-intact acrosomes stained with FITC-PNA/EthD-1 staining; (**c**) Mitochondrial membrane potential (MMP) stained with the dyes JC-1, FITC-PNA, and PI; high MMP with orange fluorescence in midpiece; and (**d**) mitochondrial membrane potential (MMP) stained; low MMP with green fluorescence in midpiece.

**Figure 2 animals-13-03040-f002:**
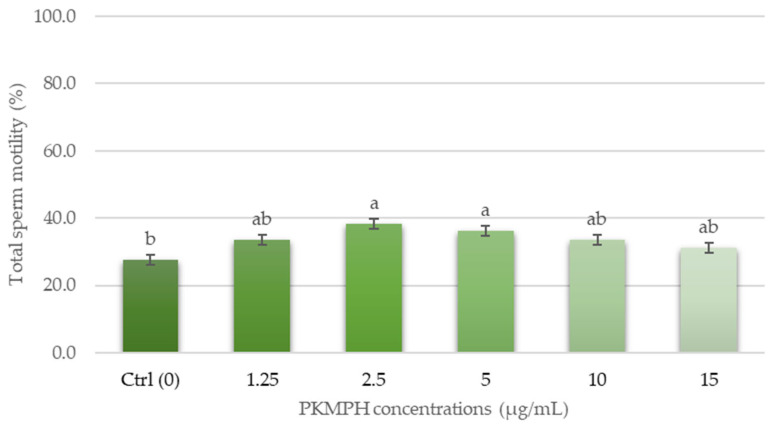
Effect of PKMPH on total sperm motility in post-thawed boar semen. Bars represent means ± SEM. Different letters indicate a statistically significant difference at *p* < 0.05.

**Figure 3 animals-13-03040-f003:**
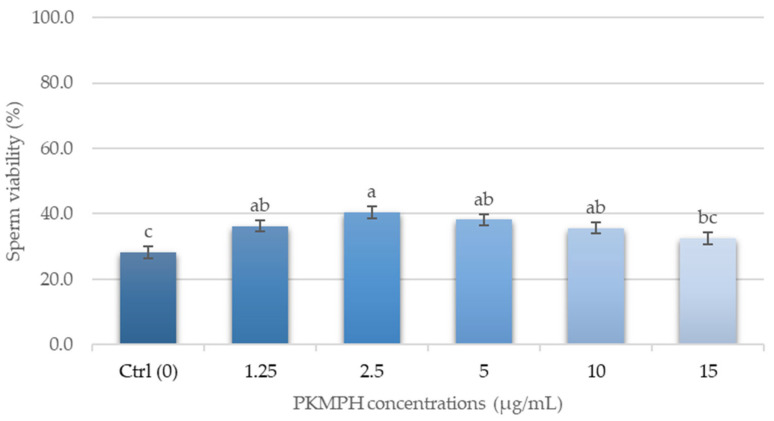
Effect of PKMPH on sperm viability in post-thawed boar semen. Bars represent means ± SEM. Different letters indicate a statistically significant difference at *p* < 0.05.

**Figure 4 animals-13-03040-f004:**
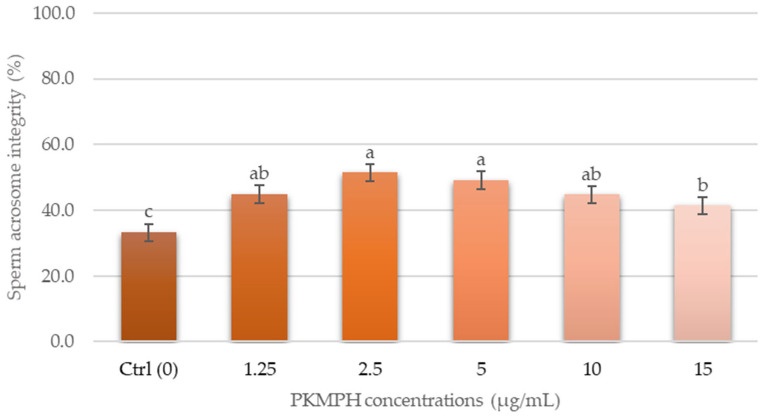
Effect of PKMPH on acrosome integrity in post-thawed boar semen. Bars represent means ± SEM. Different letters indicate a statistically significant difference at *p* < 0.05.

**Figure 5 animals-13-03040-f005:**
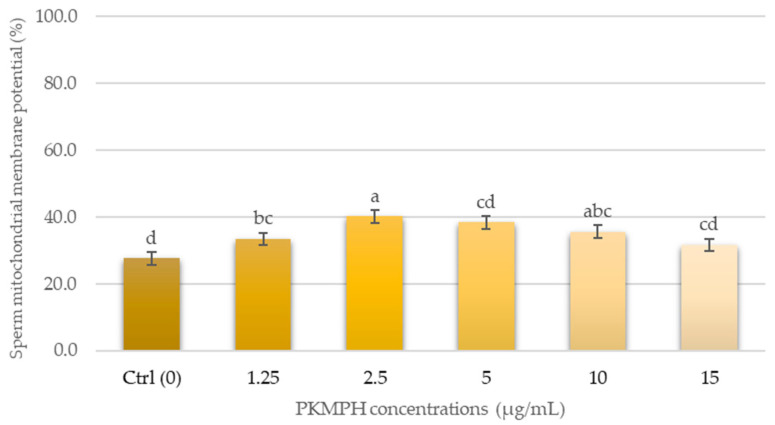
Effect of PKMPH on mitochondrial membrane potential in post-thawed boar semen. Bars represent means ± SEM. Different letters indicate a statistically significant difference at *p* < 0.05.

**Figure 6 animals-13-03040-f006:**
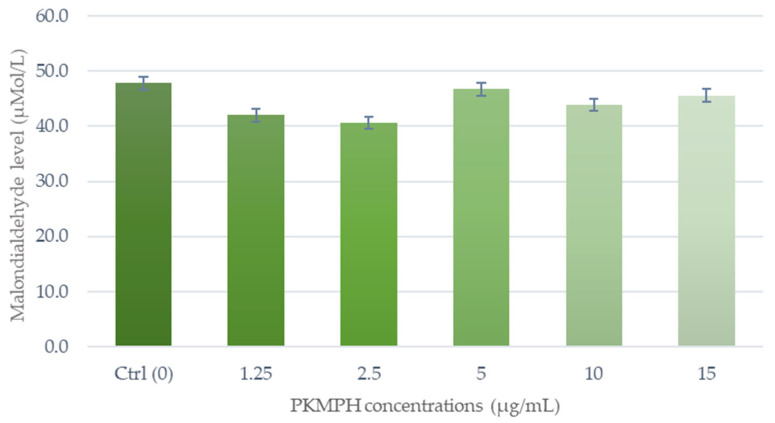
Effect of PKMPH on the lipid peroxidation (MDA level) in post-thawed boar semen. Bars represent means ± SEM. Statistically significant difference at *p* <0.05.

**Table 1 animals-13-03040-t001:** Sperm quality of fresh boar semen.

Parameters *	Mean ± SD	Range
Concentration (×10^6^ sperm/mL)	216.8 ± 57.9	110–350
Total motility (%)	93.1 ± 3.7	87–99
Progressive motility (%)	88.6 ± 5.2	82–97
Sperm viability (%)	89.1 ± 4.4	85–93
Acrosome integrity (%)	88.0 ± 4.3	82–94
Sperm morphology (%)	84.9 ± 3.6	80–90

* = Results are expressed as mean ± SEM (*n* = 17).

**Table 2 animals-13-03040-t002:** Sperm motility parameters after thawing at different PKMPH concentrations (µg/mL).

Groups *	0	1.25	2.5	5	10	15
P-MOT	18.6 ± 1.1 ^c^	22.6 ± 1.3 ^abc^	26.3 ± 2.0 ^a^	24.2 ± 1.9 ^ab^	22.3 ± 1.6 ^abc^	20.5 ± 1.7 ^bc^
VCL	29.0 ± 1.8 ^c^	37.2 ± 2.3 ^ab^	43.4 ± 2.7 ^a^	39.9 ± 2.6 ^ab^	37.4 ± 2.4 ^ab^	33.9 ± 2.1 ^bc^
VSL	10.1 ± 0.8 ^c^	13.6 ± 1.0 ^ab^	15.7 ± 1.3 ^a^	14.3 ± 1.1 ^ab^	13.5 ± 1.0 ^ab^	12.1 ± 1.0 ^bc^
VAP	12.9 ± 0.9 ^c^	17.3 ± 1.2 ^ab^	20.1 ± 1.5 ^a^	18.4 ± 1.3 ^ab^	17.3 ± 1.2 ^ab^	15.7 ± 1.2 ^bc^
ALH	0.39 ± 0.02 ^c^	0.47 ± 0.02 ^ab^	0.52 ± 0.02 ^a^	0.50 ± 0.02 ^a^	0.48 ± 0.02 ^ab^	0.44 ± 0.02 ^bc^
STR	77.1 ± 0.1 ^a^	78.2 ± 0.1 ^a^	77.4 ± 0.1 ^a^	77.1 ± 0.1 ^a^	77.8 ± 0.1 ^a^	77.3 ± 0.1 ^a^
LIN	44.0 ± 0.2 ^a^	36.1 ± 0.1 ^b^	35.2 ± 0.1 ^b^	35.5 ± 0.1 ^b^	36.7 ± 0.1 ^b^	35.1 ± 0.1 ^b^

* = Results are expressed as mean ± SEM (*n* = 17). ^a,b,c^ Means with different superscripts in the row are significantly different between groups (*p* < 0.05). P-MOT: progressive motility (%), VCL (µm/s): curvilinear velocity, VSL (µm/s): velocity straight line, VAP (µm/s): average pathway velocity, ALH (µm): amplitude of lateral head displacement, STR (%): straightness, LIN (%): linearity.

## Data Availability

No new data were created or analyzed in this study. Data sharing is not applicable to this article.

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
