# Peer review of "Palm Kernel Meal Protein Hydrolysates Enhance Post-Thawed Boar Sperm Quality"

_animals, 2023, doi:10.3390/ani13193040_

Round 1
Reviewer 1 Report
I have only a few comments and questions for the authors:
How many times was the experiment replicated?
In Semen Freezing Process, ¿the semen samples were mixed (pool) and cryopreserved as a single sample?
Line 146-147: sperm were evaluated and classified as intact acrosome or non-intact acrosome. Describe how spermatozoa with and without acrosome are observed under the fluorescent microscope.
Line 150-152: ¿Why the mitochondrial membrane potential of sperm was evaluated by fluorescent multiple staining using propidium iodide, FITC-PNA, and JC-1?
In Materials and Methods, sperm morphology is not described.
Line 190: The µg/mL is the correct unit (Figure 1).
Line 293: In conclusions, the authors concluded that palm kernel meal protein hydrolysates (PKMPH) with its bioactive peptide, is capable of reducing ROS generation during cryopreservation, inhibiting lipid peroxidation. ¿How can they confirm this if they didn't analyze the lipid peroxidation by TBARS assay (malondialdehyde (MDA) concentration) or Bodipy 581 fluorescent and ROS generation by Cellrox or DCFH-DA fluorescent?
Author Response
Dear Reviewer,
Please see an attached file for our answer. Thank you very much for your consideration.
Best regards,
Corresponding author

Reviewer 2 Report
The manuscript “Palm Kernel Meal Protein Hydrolysates Enhance Post-Thawed Boar Sperm Quality” evaluated to the effects of bioactive peptides from palm kernel meal protein hydrolysates (PKMPH) on sperm viability, motility, acrosome, and mitochondrial function in frozen–thawed boar semen. The manuscript was well written, and the methodology clearly presented. Some points are presented below:
1. My main concern is that as the authors evaluate a substance with an antioxidant action, it is necessary to have a specific analysis to verify this activity. I suggest that the authors justify this lack of analysis or insert an analysis for it. I think that the efficiency of the substance should not be associated with its antioxidant effect if no specific analysis for this activity has been carried out.
2. Abstract: I suggest that in the presentation of the results in the abstract, some numerical values are shown so that the reader can see, especially in the motility data, the antioxidant efficiency of the tested sample.
3. It is necessary to explain how the concentrations of the substance to be tested were established.
4. Material and methods: What is the control group? Would the group be cryopreserved and without the substance? Make this more clear.
5. Discussion “...elucidate the effect of PKMPH”. Rewrite this sentence, as the authors did not elucidate any effect, they just observed the effect.
6. Conclusion: “inhibiting lipid peroxidation”. What analysis was observed this event?
Author Response
Dear Reviewer,
Please see an attached file for our answer. Thank you for your consideration.
Best regards,
Corresponding author

Reviewer 3 Report
Article ID: animals-2572653
Title: Palm kernel meal protein hydrolysates enhance post-thawed boar sperm quality.
General comments
The present study tests 6 different palm kernel meal protein (PKMPH) concentrations added to the freezing extender, and its effects on sperm kinetics (by C.A.S.A.), sperm viability, acrosome integrity and mitochondrial membrane integrity. The results show that the addition of PKMPH improves all parameters compared to control. In case of mitochondrial membrane potential, the addition of 2.5 micrograms was significantly better than other amounts.
The title is adequate, and the introduction states the problem and the reasons of the study perfectly. Material and Methods is exhaustive although more insights to the tested product can be provided. Results are easy to understand and greatly exposed. The discussion and literature paragraphs are enough.
Conclusions are not completely aligned to the results and must be revised (see specific results)
Specific comments
Lines 87-89: Is PKMPH powder product a homogeneous formula? Is this experiment repeatable as it is described following the addition of x grams of that powder?
Line 93: Why were these PKMPH concentration selected? Is it based on a previous non-published experience, or literature?
Line 123: Please define the freezing curve better.
Lines 296-297: From the results it cannot be concluded that PKMPH is capable of reducing ROS generation during cryopreservation, inhibiting lipid peroxidation, because those variables were not directly measured. Please refer to the results to draw the conclusions.
Line 298: “dose” instead “does”
Lines 299-300: “quality” instead “qualities”
English language and grammar are correct.
Author Response

(The authors gave the same response as above.)

Reviewer 4 Report
Summary:
The aim of this study is to investigate the effects of bioactive peptides from palm kernel meal protein hydrolysates (PKMPH) on post-thawed boar sperm quality. The main contributions of this paper include the identification of an optimal concentration (2.5 µg/mL) of PKMPH for improving sperm motility, viability, acrosome integrity, and mitochondrial function during cryopreservation. The study highlights the potential of PKMPH as an antioxidant in preserving boar semen quality, addressing a gap in knowledge regarding the use of PKMPH in this context.
General Comments:
This paper presents valuable research on the cryopreservation of boar semen using PKMPH. It has several strengths, including a clear experimental design, robust statistical analysis, and a unique focus on PKMPH as a cryoprotectant. However, there are areas of weakness, particularly in the discussion of the antioxidant mechanism of PKMPH and the cytotoxic effects observed at higher concentrations. Additionally, the review of relevant literature could be more comprehensive to provide context for the study.
Specific Comments:
1. Specify the criteria used for selecting the 2.5 µg/mL concentration of PKMPH as optimal. Was this based solely on post-thawed sperm quality, or were other factors considered?
2. Provide more details on the antioxidant mechanisms of PKMPH, including how it neutralizes ROS and reduces lipid peroxidation. This would enhance the understanding of the observed effects.
3. Mention the potential practical implications of PKMPH supplementation for artificial insemination or other applications in the boar breeding industry.
4. Clarify whether the variation in results among individual boars is discussed in the paper and its potential impact on practical applications.
5. Ensure consistency in units; here, µg/mL is mentioned, but in Table 1, mg/mL is used. Confirm the correct unit.
6. Discuss the limitations of the study, including the potential cytotoxic effects observed at higher PKMPH concentrations, and their implications for practical use.
7. Suggest avenues for future research, such as investigating the specific antioxidant pathways activated by PKMPH.
Methodology:
1. Study Design: The study design appears appropriate, involving a controlled experiment with different concentrations of palm kernel meal protein hydrolysates (PKMPH) in the freezing extender to assess their impact on post-thawed boar sperm quality.
2. Sample Size and Randomization: The use of 17 boar semen ejaculates is mentioned, but there's no explanation for the choice of this sample size. It would be helpful to provide a rationale for sample size determination and how the samples were randomized.
3. Control Group: The methodology mentions a control group but doesn't specify what this group comprises. Clarification on the exact composition of the control group and its role in the study would be beneficial.
4. Data Collection: The methods for collecting semen samples, preparing freezing extenders, cryopreserving semen, and thawing are well-detailed. However, it would be helpful to include details such as the temperature and duration of storage in liquid nitrogen.
Statistics:
1. Statistical Analysis: The statistical analysis is appropriately described, including the use of ANOVA and Duncan's multiple range test. This provides a clear understanding of how data were analyzed.
2. Significance Levels: The paper mentions that a p-value less than 0.05 was considered statistically significant, which is a standard practice. However, it would be beneficial to state this explicitly in the statistical analysis section.
Results:
1. Data Presentation: The presentation of results is clear and supported by tables and figures, making it easy to understand the effects of different concentrations of PKMPH on sperm quality.
2. Interpretation: The results are discussed in detail, including comparisons between treatment groups and the control group. The authors correctly interpret the findings and relate them to the study's objectives.
Conclusions:
1. Key Findings: The conclusions effectively summarize the key findings of the study, emphasizing the positive impact of PKMPH on post-thawed boar sperm quality.
2. Optimal Concentration: The paper identifies an optimal concentration of 2.5 µg/mL of PKMPH in the freezing extender for improving sperm quality. This information is valuable for practical applications.
3. Limitations: It's important to acknowledge the limitations of the study. For instance, the cytotoxic effects observed at higher PKMPH concentrations should be discussed in more detail. While the paper does not delve deeply into the underlying antioxidant mechanisms of PKPH, it hints at its potential antioxidant effects on sperm. Further research into the specific mechanisms could contribute to the novelty of the findings.
General Suggestions:
1. Clarity: While the paper is generally well-written, some sentences are lengthy and complex. Consider breaking down certain statements into smaller, more concise sentences to enhance readability.
2. Inclusion of Antioxidant Mechanism: Discuss the potential antioxidant mechanisms of PKMPH in protecting sperm during cryopreservation. Explain how PKMPH reduces ROS and lipid peroxidation.
3. Sample Variation: Acknowledge and discuss the observed variation in results among individual boars, as this could impact the practical application of the findings.
4. Future Research: Suggest avenues for future research, such as investigating the molecular mechanisms underlying PKMPH's antioxidant effects on sperm.
Overall, the paper provides valuable insights into the use of PKMPH to improve post-thawed boar sperm quality. Addressing the suggestions above can enhance the clarity and comprehensiveness of the paper.
The overall quality of English language usage in the paper appears to be good. The text is generally well-written and coherent, making it easy to understand. However, there are some areas where improvements can be made for clarity and precision:
1. Sentence Structure: The sentence structure is mostly clear and concise, but there are a few sentences that are quite long and complex. It would be beneficial to break down these sentences into shorter ones to improve readability.
- Line 201:"Supplemented with 2.5 µg/mL, it showed the highest number of viable sperm (40.5 ± 1.8%) and was higher than control by 12.3%."
- Recommendation: "At a concentration of 2.5 µg/mL supplementation, it demonstrated the highest number of viable sperm (40.5 ± 1.8%), which was 12.3% higher than the control."
2. Grammar and Syntax: The paper generally follows proper grammar and syntax rules, but there are instances where sentence construction could be refined for better clarity.
Line 73-74: "However, no study has been reported on 73 the antioxidant effect of PKMPH during cryopreservation of boar semen."
-Recommendation: "However, there have been no studies reported on the antioxidant effects of PKMPH during the cryopreservation of boar semen."
3. Terminology: The use of scientific terminology is appropriate, but there are a few terms and abbreviations that could be defined or explained upon first use to ensure clarity for readers who may not be experts in the field.
4. Units of Measurement: Ensure consistency in the use of units of measurement. For example, in Table 1, both µg/mL and mg/mL are used, and it's important to use the correct unit consistently throughout the paper.
5. Punctuation: Check for consistent and proper use of punctuation marks, especially in lists and series of items.
Overall, while the paper is well-written, a thorough proofreading for language and grammar, along with addressing the specific points mentioned, will help enhance the quality of English language usage in the manuscript.
Author Response

(The authors gave the same response as above.)

Round 2
Reviewer 1 Report
Comments and Suggestions for Authors
It is recommend to make some changes to the paragraph (Lines: 117-120). For exampple, The straws were placed in contact with nitrogen vapor (4 cm above the leve lof the liquid nitrogen) for 20 min (-20°C /min) in a polystyrene box and plunged into the liquid nitrogen tank (-196°C) for storage prior to análisis.
Author Response
Line 117-120, the sentences have been changed according to reviewer suggestions and marked with RED as follows: The straws were placed in contact with nitrogen vapor (4 cm above the level of the liquid nitrogen) for 20 min (-20°C /min) in a polystyrene box and plunged into the liquid nitrogen tank (-196°C) for storage prior to analysis.
Reviewer 2 Report
The manuscript was revised.
Author Response
Thank you very much.
Reviewer 3 Report
All my questions and doubts were satisfactory solved.
Author Response
Thank you very much.